# Early Fusion of H&E and IHC Histology Images for Pediatric Brain Tumor Classification

**Christoforos Spyretos**[1,2]                              CHRISTOFOROS.SPYRETOS@LIU.SE
**Iulian Emil Tampu**[1,2]                                   IULIAN.EMIL.TAMPU@LIU.SE
**Nadieh Khalili**[3]                                  NADIEH.KHALILI@RADBOUDUMC.NL
**Juan Manuel Pardo Ladino**[1,4]                  JUAN.MANUEL.PARDO.LADINO@LIU.SE
**Per Nyman**[2,5]                                 PER.NYMAN@REGIONOSTERGOTLAND.SE
**Ida Blystad**[2,6]                                 IDA.BLYSTAD@REGIONOSTERGOTLAND.SE
**Anders Eklund**[1,2,4]                                         ANDERS.EKLUND@LIU.SE
**Neda Haj-Hosseini**[1,2]                                 NEDA.HAJ.HOSSEINI@LIU.SE

*1 Department of Biomedical Engineering, Linköping University, Sweden*

*2 Center for Medical Image Science and Visualization, Linköping University, Sweden*

*3 Department of Pathology, Radboud University Medical Center, The Netherlands*

*4 Department of Computer and Information Science, Linköping University, Sweden*

*5 Crown Princess Victoria Children's Hospital and Department of Health, Medicine and Caring Sciences, Linköping University, Linköping, Sweden*

*6 Department of Radiology and Department of Health, Medicine and Caring Sciences, Linköping University, Sweden*

## Abstract

This study explores the application of computational pathology to analyze pediatric brain tumors utilizing hematoxylin and eosin (H&E) and immunohistochemistry (IHC) whole slide images (WSIs). Experiments were conducted on H&E images for predicting tumor diagnosis and fusing them with unregistered IHC images to investigate potential improvements. Patch features were extracted using UNI, a vision transformer (ViT) model trained on H&E data, and whole slide classification was achieved using the attention-based multiple instance learning CLAM framework. In the astrocytoma tumor classification, early fusion of the H&E and IHC significantly improved the differentiation between tumor grades (balanced accuracy: $0.82 \pm 0.05$ vs $0.84 \pm 0.05$). In the multiclass classification, H&E images alone had a balanced accuracy of $0.79 \pm 0.03$ without any improvement obtained when fused with IHC. The findings highlight the potential of using multi-stain fusion to advance the diagnosis of pediatric brain tumors, however, further fusion methods should be investigated.

**Keywords:** pediatric brain tumour, H&E, immunohistochemistry (IHC), Ki-67, GFAP, computational pathology, early fusion, UNI, CLAM, foundation model

## 1 Introduction

Central nervous system (CNS) tumors were the second leading cause of cancer incidences among children and adolescents aged 0-19 years old, with worldwide incidence and mortality rates of 1.2 and 0.6 (per 100,000 people) in 2022 (Ferlay J, 2024). WSIs assist pathologists in diagnosing brain tumors, providing a cost-effective illustration, sharing, and archiving pathology information. The introduction of WSIs has led to an extensive volume of data, facilitating the implementation of deep learning in assisting pathologists with making faster and more consistent decisions.

The world health organization's (WHO) latest edition of guidelines for the CNS tumors classification published in 2021, emphasizes the integration of molecular information into diagnosing brain tumors (Louis et al., 2016). Despite the recent advancements in molecular diagnostics,

the assessment of histology remains an essential element in evaluating CNS tumors (Viaene, 2023) and determining the specific ancillary molecular tests needed for diagnosis. Pathologists often use various immunohistochemical (IHC) stains in addition to H&E to identify specific molecular alterations, detect mutant proteins, and determine the molecular subgroups of brain tumors.

The increasing availability of biomedical data, such as medical imaging, electronic health records and genome sequences, has led to the development of multimodal artificial intelligence applications (Kline et al., 2022; Acosta et al., 2022). These applications mimic the multimodal nature of clinical expert decision-making, aiming to enhance predictions and achieve more accurate diagnoses. Few studies in computational pathology have analyzed H&E stained WSIs alongside IHC stained WSIs, and those available are on breast cancer (Weitz et al., 2021; Liu et al., 2020). In this study, the classification of pediatric brain tumors is investigated using H&E stained images. In addition, it is examined whether the early fusion of unregistered H&E, Ki-67, and glial fibrillary acidic protein (GFAP) stained images improves the diagnostic predictions compared to only using H&E slides. To the best of our knowledge, no published study has yet explored the potential of fusing registered or unregistered H&E and IHC images for predicting pediatric or adult brain tumor diagnoses. The Ki-67 immunostaining is interpreted using the Ki-67 labeling index (Ki-67 LI) and is defined as the percentage of the number of Ki-67 positive tumor nuclei (brown cells) divided by the number of all tumor nuclei, correlating with the histological tumor grade. Potential uses for Ki-67 include diagnosis of medulloblastomas and astrocytomas and distinguishing between astrocytoma grades (Sengupta et al., 2012; Sharma et al., 2018). GFAP is a reliable marker for histological diagnosis between glial and non-glial tumors and the grading of astrocytomas (Varma et al., 2018; van Bodegraven et al., 2019).

## 2 Data

In this study, WSIs were utilized from the children's brain tumor network (CBTN) dataset, which consists of over 2,000 subjects and more than 8,000 slides (Lilly et al., 2023; Shapiro et al., 2023). The slides have a $\times 20$ magnification with a pixel resolution ranging from approximately 0.251 to 0.505 micrometers per pixel. However, the CBTN dataset is based on pre-2021 WHO guidelines, which include brain tumor classifications no longer used in clinical practice. Therefore, subjects with outdated tumor classifications were excluded, and the brain tumor classifications listed in the 2021 WHO guidelines were used to conduct the analysis.

Most WSIs are H&E-stained, and the most represented IHC stains are Ki-67 and GFAP. Thus, only subjects with slides containing all three stains were selected to conduct the experiments. It is important to note that the slides are unregistered, and subjects might contain several single-modality WSIs. In addition, a threshold of 10 subjects per class was set to ensure a sufficient representation of each tumor family/type. Consequently, the study analyzes the cancer types of ependymoma (EP), medulloblastoma (MED), and ganglioglioma (GANG), and the tumor families of astrocytoma low-grade glioma (grades 1, 2) (ASTR-LGG) and astrocytoma high-grade glioma (grades 3, 4) (ASTR-HGG). ASTR-LGG, ASTR-HGG, GANG, and EP are glial tumors, while MED is a non-glial tumor. Additionally, images with artifacts such as pen marks and air bubbles were included in the experiments, as their effect was negligible in the model's performance (Pardo Ladino, 2024). Table 1 summarizes the number of subjects and WSIs included in the study, and figure 1 is an example from the dataset.

## 3 Methodology

End-to-end classification of WSIs using deep learning is challenging due to their immense size, typically in the range of gigapixels (Hosseini et al., 2024). This limitation is commonly circumvented by training patch-based networks using pixel-level annotations and then aggregating the patch-level results. Those approaches are known as supervised learning methods. However,

Table 1: Tumor families/types and the number of subjects, H&E, Ki-67, and GFAP slides used in the analysis. Only subjects with slides containing all three stain modalities were included in the experiments.

| Tumor family/type | Number of subjects | Number of H&E slides | Number of Ki-67 slides | Number of GFAP slides |
|---|---|---|---|---|
| ASTR-LGG | 173 | 342 | 195 | 196 |
| ASTR-HGG | 64 | 127 | 71 | 76 |
| EP | 47 | 106 | 54 | 57 |
| MED | 46 | 80 | 52 | 51 |
| GANG | 40 | 91 | 47 | 52 |
| Totals | 370 | 746 | 419 | 432 |

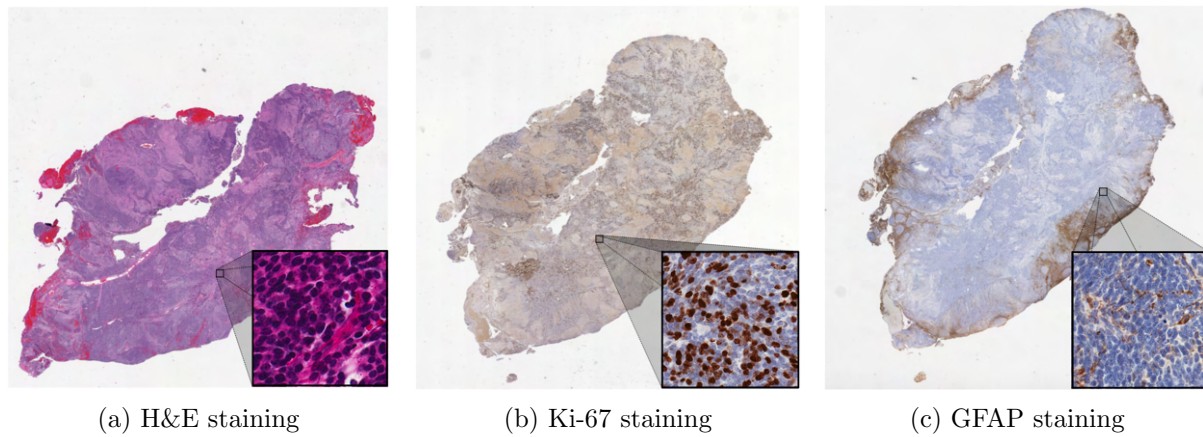

(a) H&E staining      (b) Ki-67 staining      (c) GFAP staining

Figure 1: Examples of H&E, Ki-67, and GFAP WSIs from the same subject diagnosed with medulloblastoma.

obtaining a large amount of patches with fine-grained annotations is expensive, time-consuming and requires the input of skilled and experienced pathologists. Therefore, weakly-supervised and self-supervised learning methods have been developed in computational pathology, addressing the aforementioned issues. Under the weakly-supervised learning paradigm, models are trained on partially or sparsely labeled data, such as one label for an entire WSI or per subject. With self-supervised learning, the algorithms learn feature representations through unlabeled data.

Multiple instance learning (MIL) is a weakly supervised learning method, where each WSI is considered as a bag containing multiple patches, also called instances (Carbonneau et al., 2018). If a WSI (bag) is labeled class-positive, then at least one patch (instance) from that WSI is class-positive. Otherwise, if a WSI is class-negative, all patches belonging to that WSI are negative. In this study, a MIL approach named clustering-constrained attention multiple instance learning (CLAM) was utilized to perform the classification tasks (Lu et al., 2021), aggregating patch-level features in slide-level representations for classification.

In the pre-processing phase of the images, the CLAM toolbox was employed to segment the tissue and extract patches of 256×256 pixels and their features. The ViT named UNI (UNI-ViT) was utilized as a feature extractor, pre-trained on a proprietary histology dataset (Chen et al., 2024a). UNI is a recently introduced foundation model pre-trained on more than 100 million patches from over 100,000 diagnostic in-house H&E-stained WSIs across 20 major tissue types through the self-supervised DINOv2 framework (Oquab et al., 2023). Feature extractors pre-trained on in-domain histology datasets can capture a broad spectrum of patterns, such as different fixations, staining characteristics, scanning protocols, and tissue architecture across

multiple centers. After extracting the features, those corresponding to the same stain modality for each subject were concatenated, representing subject-level features per stain modality. An early fusion approach was utilized to explore the potential improvement in diagnostic prediction through the integration of IHC with H&E staining. Specifically, all possible fused combinations between the three stain modalities were investigated by concatenating features at the subject level. Figure 2 illustrates the workflow for the single-stain modality and multi-stain approaches.

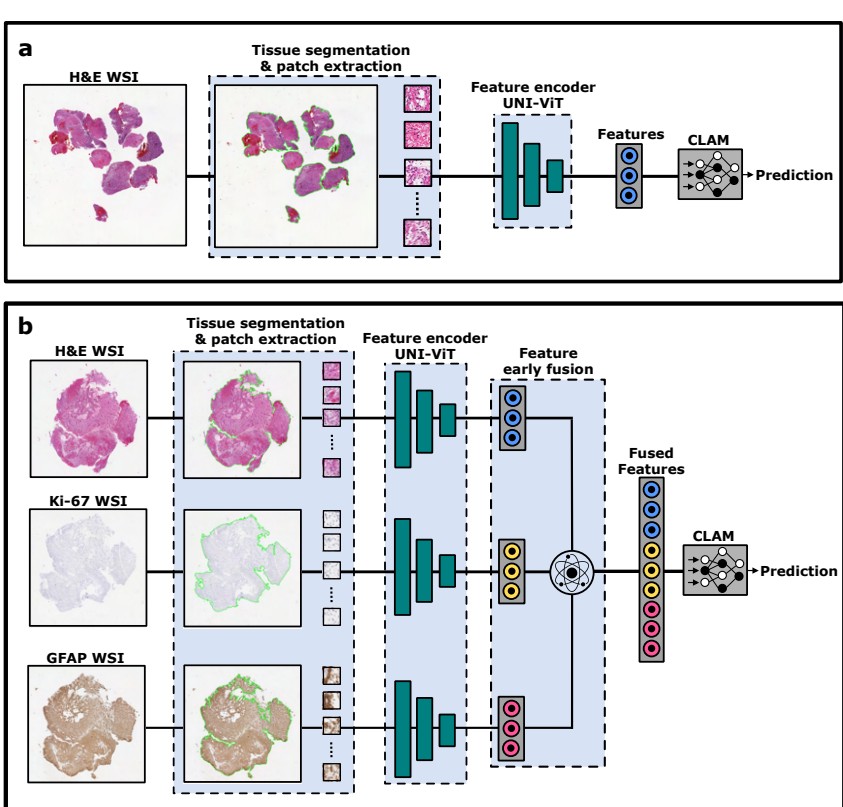

Figure 2: Overview of the **a)** single-stain modality and **b)** multi-stain modalities workflows. **a)** Tissue segmentation and patch extraction from the WSIs were accomplished using the CLAM toolbox. The extracted patches were encoded to feature representations using UNI, a ViT pretrained on an in-domain proprietary H&E histology dataset. These features were given as input into the CLAM model to perform the classification tasks. **b)** Tissue segmentation, patch and feature extraction were conducted in the same manner as in the single-stain modality procedure for each stain. An early fusion approach was employed between all possible combinations of the three stain modalities by concatenating features at the subject level. The fused features were then fed into the CLAM model to perform the classification tasks. Figure inspired from (Lipkova et al., 2022)

The classification tasks were performed using the small-sized single-branch CLAM model. In the training phase, the learning rate was 1e-4, and the maximum and minimum number of epochs was set at 20 and 10, respectively. Early stopping was utilized on the validation loss with patience set to 5 epochs. All other settings were set at the default values suggested by (Lu et al., 2021). In addition to the CLAM framework, class weight adjustment was used to mitigate the imbalanced data distribution, using the Scikit-learn library (Pedregosa et al., 2011), with weights being inversely proportional to the number of samples in each class.

The dataset was class-stratified and subject-wise split into 50% for training, 20% for validation, and 30% for testing to prevent data leakage between the sets. Since the class distribution of the dataset is imbalanced, the models were assessed using balanced accuracy, Matthew's correlation coefficient (MCC), area under the receiver operating characteristic curve (AUC-ROC)

and weighted F1-score. Experiments were conducted on individual stain modalities and all combinations of the fused stain modalities using non-parametric bootstrapping with 50 replicates. Non-parametric bootstrapping involves repeatedly sampling from the dataset with replacement to generate multiple training, validation, and test sets (replicates). This approach allows for robust estimations of statistical measures without making assumptions about the data distribution. For the statistical comparison, 10,000 permutations were conducted between the test sets to assess observed differences in the performance of the models, at a significance level of $\alpha = 0.05$. Statistical comparisons were conducted between the H&E single-stain modality model and the fused stain modality models, as well as comparisons between the fused models. Bonferroni correction was utilized to adjust the significance level, and each hypothesis was performed at a statistical significance level $\alpha = 0.05/6 \approx 0.0083$ (Napierala, 2012).

Attention maps were used to visualize and interpret the importance of regions in the WSI, utilizing the CLAM toolbox. Furthermore, QuPath (Bankhead et al., 2017) was employed to generate the positive and negative cell density maps of the Ki-67 using settings proposed by (Pai et al., 2022), which were visually compared to the attention maps, and to calculate the Ki-67 LI for the WSI.

## 4 Results

Experiments were conducted using the three different staining modalities individually and in all possible combinations. Table 2 illustrates the classification performance between ASTR-LGG and ASTR-HGG and between the five tumor families/types. The models were assessed using unseen test sets and tables show the mean values of the metrics along with their standard deviation across 50 repetitions.

In binary classification of ASTR-LGG and ASTR-HGG, the fusion of all three stains achieves the highest performance, with significant differences compared to using only the H&E WSIs in most metrics, although it is not always significantly better than fusing H&E WSIs with either Ki-67 or GFAP WSIs only. The performance of each class steadily improved by gradually combining different stain modalities. For ASTR-LGG, the balanced accuracy increased from $0.91 \pm 0.04$ to $0.92 \pm 0.05$, and the F1-score from $0.73 \pm 0.07$ to $0.77 \pm 0.06$ when fusing three stains. Similarly, for ASTR-HGG, the balanced accuracy improved from $0.72 \pm 0.1$ to $0.75 \pm 0.1$, and the F1-score from $0.91 \pm 0.02$ to $0.92 \pm 0.02$ when fusing the three stains. However, in the five-class classification, fusing Ki-67, GFAP, or both with H&E does not significantly improve the performance compared to using H&E WSIs alone, and there is no improvement in each class's performance. Specifically, using H&E WSIs alone, MED (F1-score: $0.91 \pm 0.04$) and EP (F1-score: $0.90 \pm 0.05$) are almost perfectly classified, and ASTR-LGG (F1-score: $0.83 \pm 0.03$) is correctly classified in most cases. However, ASTR-HGG (F1-score: $0.71 \pm 0.08$) and GANG (F1-score: $0.57 \pm 0.11$) are frequently misclassified as ASTR-LGG.

Figure 3 illustrates the attention maps for individual and fused stain modalities. Red regions indicate higher attention from the model, while blue regions indicate lower attention in those tissue regions. Figures 4 and 5 illustrate the attention and cell density maps of an ASTR-HGG Ki-67 WSI and ASTR-LGG Ki-67 WSI, respectively. The positive cell density map shows Ki-67 positive stained nuclei (brown cells) and the negative cell density map shows Ki-67 negatively stained nuclei (blue cells); the greater the ratio of positive cells to negative cells, the higher the tumor grade. The Ki-67 LI of the ASTR-HGG is 3.6%, suggesting a high proliferation rate associated with high-grade pediatric brain tumors. By comparing the attention map to the positive cell density map, it could be seen that the model's attention is localized in the WSI's regions that have a high density of positive Ki-67 (brown) cells (red areas). In contrast, the Ki-67 LI of the ASTR-LGG is 0.4%, indicating a low proliferation rate associated with low-grade pediatric brain tumors. When comparing the attention map to the negative cell density map, it could be interpreted that the model's attention is mainly localized in the upper left

Table 2: Classification performances between ASTR-LGG and ASTR-HGG and five-class classification of tumor families/types using the three different staining modalities, individually and in all possible combinations, across 50 repetitions. The models with the best overall performance are highlighted in bold.

| Stain Modalities | ASTR-LGG vs ASTR-HGG | | | |
| --- | --- | --- | --- | --- |
| | Balanced Accuracy | MCC | AUC-ROC | Weighted F1-Score |
| H&E | $0.82 \pm 0.05^a$ | $0.65 \pm 0.08^a$ | $0.90 \pm 0.04^a$ | $0.86 \pm 0.03^a$ |
| H&E + Ki-67 | $0.83 \pm 0.05^b$ | $0.68 \pm 0.09^b$ | $0.90 \pm 0.03^b$ | $0.87 \pm 0.03^b$ |
| H&E + GFAP | $0.82 \pm 0.05^c$ | $0.66 \pm 0.09^c$ | $0.91 \pm 0.04^c$ | $0.87 \pm 0.04^c$ |
| **H&E + Ki-67 + GFAP** | $\mathbf{0.84 \pm 0.05^a}$ | $\mathbf{0.69 \pm 0.08^a}$ | $\mathbf{0.90 \pm 0.04^d}$ | $\mathbf{0.88 \pm 0.03^a}$ |
| Ki-67 | $0.81 \pm 0.04$ | $0.62 \pm 0.07$ | $0.88 \pm 0.05$ | $0.85 \pm 0.03$ |
| GFAP | $0.77 \pm 0.05$ | $0.55 \pm 0.08$ | $0.86 \pm 0.04$ | $0.82 \pm 0.03$ |
| Five-class classification between tumour families/types | | | | |
| **H&E** | $\mathbf{0.79 \pm 0.03^a}$ | $\mathbf{0.72 \pm 0.04^a}$ | $\mathbf{0.94 \pm 0.01^a}$ | $\mathbf{0.80 \pm 0.03^a}$ |
| H&E + Ki-67 | $0.78 \pm 0.03^b$ | $0.73 \pm 0.04^b$ | $0.94 \pm 0.01^{a,b}$ | $0.80 \pm 0.03^b$ |
| H&E + GFAP | $0.78 \pm 0.04^c$ | $0.72 \pm 0.05^c$ | $0.94 \pm 0.01^b$ | $0.80 \pm 0.03^c$ |
| H&E + Ki-67 + GFAP | $0.78 \pm 0.03^d$ | $0.72 \pm 0.05^d$ | $0.94 \pm 0.01^c$ | $0.80 \pm 0.03^d$ |
| Ki-67 | $0.66 \pm 0.04$ | $0.58 \pm 0.05$ | $0.88 \pm 0.02$ | $0.70 \pm 0.04$ |
| GFAP | $0.69 \pm 0.05$ | $0.61 \pm 0.05$ | $0.90 \pm 0.02$ | $0.72 \pm 0.04$ |

$^{a,b,c,d}$ Within a column, two-sided p-value $< 0.05$ permutation test, significance level adjusted after Bonferroni correction to $\alpha = 0.05/6 \approx 0.0083$. Model performances with a common superscript differ significantly.

part of the tissue with a high negative cell (blue) region (red area) according to the cell density map. However, the model's attention is also concentrated on tissue regions with a low density of negative cells.

## 5 Discussion

Healthcare professionals utilize various information sources to make comprehensive decisions and a thorough understanding of the patient's condition. By fusing multiple stain modalities, different tissue characteristics can be obtained through diverse stains which combined could lead to a more comprehensive and accurate analysis (Lipkova et al., 2022). The experiments conducted using unregistered WSIs with three different staining modalities (H&E, Ki-67, and GFAP), both individually and fused, produced satisfactory results in binary and five-class classifications. When distinguishing between ASTR-LGG and ASTR-HGG, fusing all three stains significantly outperformed H&E slides alone. This result is in agreement with the utilization of Ki-67 and GFAP in the clinic and highlights the potential diagnostic value of Ki-67 and GFAP in differentiating between astrocytoma grades in deep learning-based analysis. However, in the five-class classification task, although the performance of the model on the H&E WSIs alone was high, fusing Ki-67 WSIs, GFAP WSIs, or both with H&E WSIs did not improve the results, implying that the IHC information is not useful for multi-class tumor type/family classification due to undistinguished IHC information in the multiple tumor classes or that a different fusion strategy is needed.

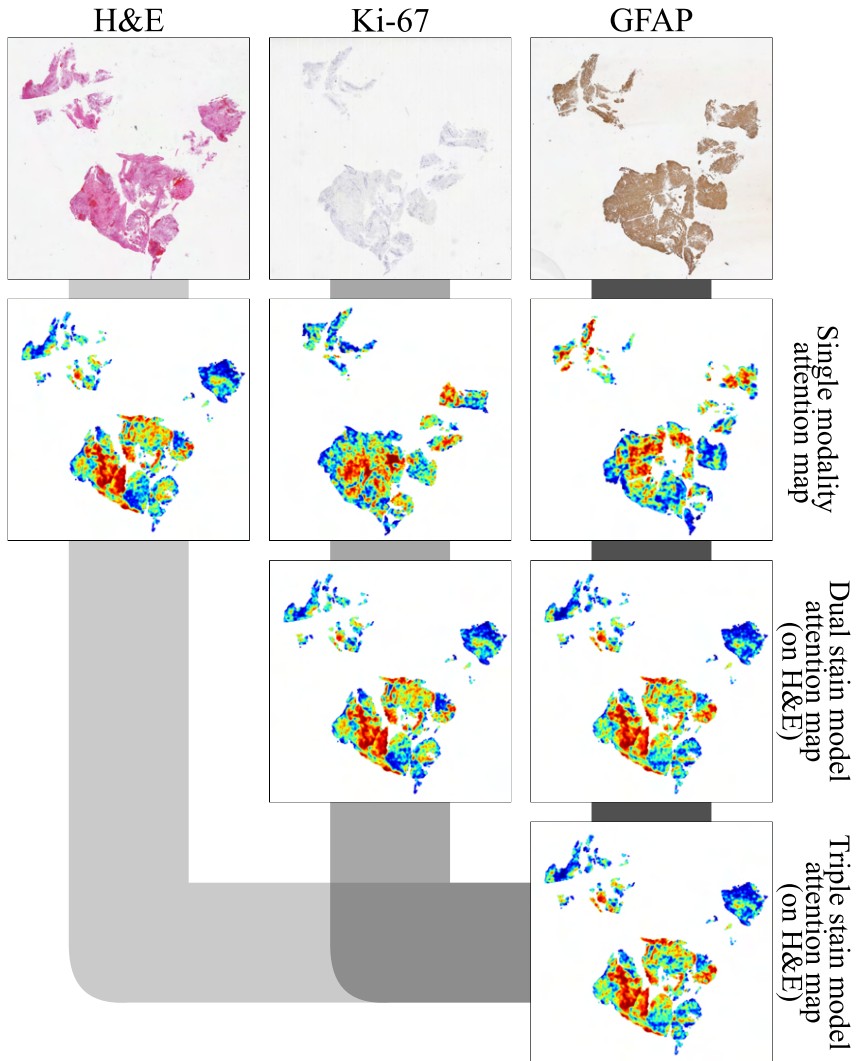

Figure 3: Attention maps of ASTR-LGG WSIs of the ASTR-LGG vs ASTR-HGG classification task between individual and fused stain modalities.

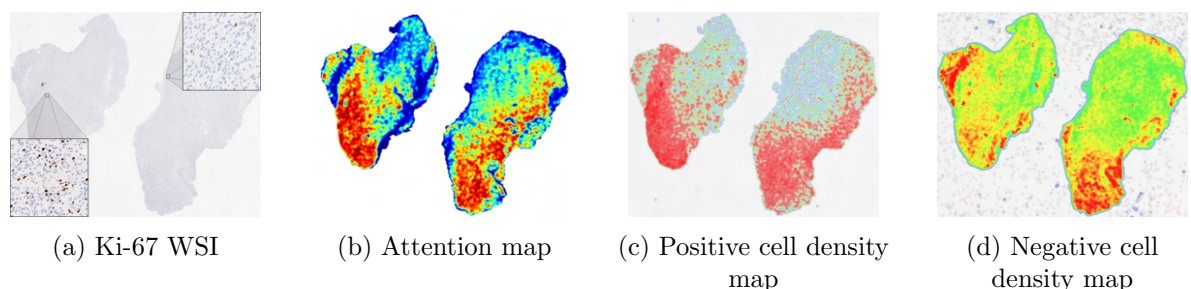

(a) Ki-67 WSI      (b) Attention map      (c) Positive cell density map      (d) Negative cell density map

Figure 4: Ki-67 WSI of an ASTR-HGG with a Ki-67 LI of 3.6%, alongside corresponding attention and cell density maps.

The comparison between the attention and the cell density maps provides a useful understanding of how the model's attention is localized to the regions of the tissue based on cell proliferation activity. In low-grade tumors, the model emphasizes tissue areas with negatively stained nuclei, whereas in high-grade tumors, the model's attention is mainly concentrated on regions with high Ki-67 positively stained nuclei, suggesting the model's ability to classify tu-

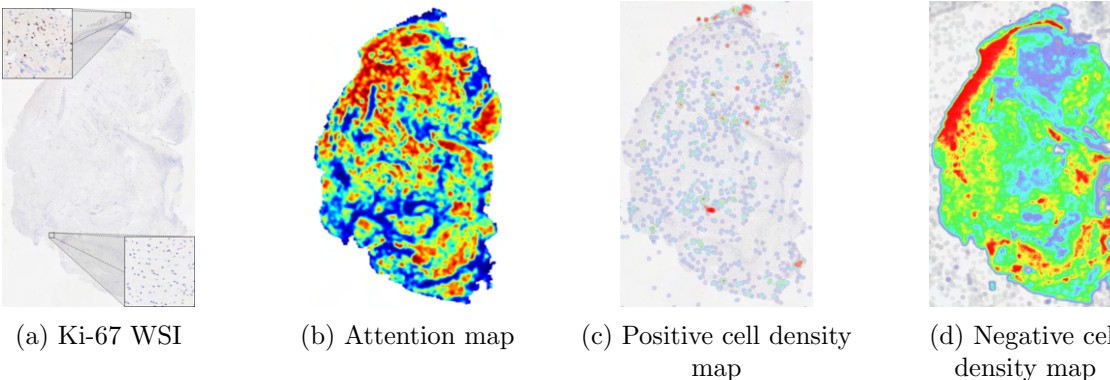

| (a) Ki-67 WSI | (b) Attention map | (c) Positive cell density map | (d) Negative cell density map |

Figure 5: Ki-67 WSI of an ASTR-LGG with a Ki-67 LI of 0.4%, alongside corresponding attention and cell density maps.

mor grades based on cellular proliferation markers. However, a more thorough analysis would require further interpretation of the maps and eventual statistical analysis.

To further explore the potential of multi-stain fusion in computational pathology, additional fusion strategies beyond early fusion should be investigated. Multimodal strategies also include late and intermediate fusions. In addition, given the development of MIL models in digital pathology, evaluating other slide aggregation methods should be considered (Chen et al., 2024b) among which ABMIL can potentially improve predictions on H&E images (Tampu et al., 2024). Moreover, UNI-ViT was utilized as the feature extractor since it is among the most recent and best-performing histology foundation models. However, it should be mentioned that newly published histology foundation models, such as Virchow (Vorontsov et al., 2024), might be more effective feature extractors given the comparable performance on adult brain tumors. Furthermore, only WSIs were used for model training, although fusing molecular data should also be considered, potentially leading to a more improved and refined diagnosis; in accordance with the 2021 WHO guidelines for CNS tumors that underline the importance of molecular information in brain tumor diagnosis. Recent studies have concluded that fusing WSIs with molecular data improves the performance of various downstream tasks in adult brain tumors (Chen et al., 2022; Wang et al., 2023; Xing et al., 2022; Pei et al., 2021). For pediatric brain tumors, Steyart et al. demonstrated that fusing WSIs with gene expression profiles resulted in more accurate prognosis predictions compared to using WSIs alone (Steyaert et al., 2023).

## 6 Conclusion

The experiments showed promising results in classifying pediatric brain tumor families/types using H&E slides. Early fusion of unregistered Ki-67 and GFAP with H&E slides significantly improved the distinction between ASTR-LGG and ASTR-HGG, suggesting the diagnostic potential of Ki-67 and GFAP stains. However, over multiple tumor classes, fusion of the H&E and IHC did not improve the results from what is obtained by the H&E images only. Alternative fusion strategies beyond early fusion, such as late and intermediate fusions, should be explored in the next step to maximize the potential of multi-stain fusion for this application.

## Acknowledgements and Disclosure of Funding

The research was made possible in part due to The Children's Brain Tumor Tissue Consortium (CBTTC)/ The children's brain tumor network (CBTN). The study was financed by Swedish Childhood Cancer Foundation (MT2021-0011, MT2022-0013), Joanna Cocozza's Foundation (2023-2024), Linköping University's Cancer Strength Area (2022) and Vinnova project 2017-02447 via Medtech4Health and Analytic Imaging Diagnostics Arena (1908), no. 2222.

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
