# OpenReview forum: "Early Fusion of H&E and IHC Histology Images for Pediatric Brain Tumor Classification"
_MICCAI.org/2024/Workshop/COMPAYL — COMPAYL 2024_

### Official Review · Reviewer_MQkV · 2024-07-10
**Review of Early Fusion of H&E and IHC Histology Images for Pediatric  Brain Tumor Classification**

**Custom Rating:** 4
**Confidence:** 4

**Review:**

The study investigates the early fusion of unregistered H&E, Ki-67, and glial fibrillary acidic protein (GFAP) stained images for pediatric brain tumor classification, comparing it to using only H&E images. Patch features were extracted using UNI, and whole slide classification was performed using the CLAM framework. In astrocytoma tumor classification, the early fusion of H&E and IHC significantly improved differentiation between tumor grades.

Pros:
- The authors use UNI, a novel foundation model for feature extraction.
- They also employed four metrics for robust quantitative evaluation (balanced accuracy, MCC, AUC ROC, and weighted F1-score).
- Conducts experiments using the three different staining modalities individually and in all possible combinations.
- Fusion of all three stains achieves the highest performance for the binary task between ASTR-LGG and ASTR-HGG in almost all metrics.
- The paper shows a preliminary interpretability analysis using heatmaps of the model attention and correlation with Ki-67 positive and negative cell density maps.
- According to the authors, no published study has yet explored the potential and feasibility of fusing unregistered H&E and IHC images for predicting pediatric or adult brain tumor diagnoses.

Cons:
- The proposed method does not show a general improvement; fusing Ki-67, GFAP, or both with H&E did not improve the five-class
 classification task compared to using only H&E images.
- The study exclusively uses CLAM. Given the development of multiple instance learning (MIL) models in digital pathology, evaluation with other MIL models should be considered.

Minor comments
-        Typo on page 2: “These applications mimic the multimodal nature of clinical expert decision-making and aim to enhance predictions and achieve more accurate diagnoses. enhance predictions and achieve more accurate diagnoses.”

---

### Official Review · Reviewer_wVJb · 2024-07-15
**This paper shows an interesting approach for integrating multiple stainings for a more accurate classification of brain tumors in children with promising results.**

**Custom Rating:** 4
**Confidence:** 3

**Review:**

Strengths:
- Clear motivation and a good introduction
- Integration of multiple special staining with H&E (also containing genomics data)
- All combinations tested of fusion

Weaknesses/room for improvement:
- It is a bit misleading to say that the text that the dataset is very big when in fact only about 10-20% of the data is used (2. Data)
- Reference on UNI-ViT as it is apparently recently introduced)?
- Fig. 2 seems to have sub-optimal image resolution and appears pixelated
- Table 2: in the AUC-ROC column for the 5-class set-up it is indicated that the model with the same reported performance has a statistically significant difference, how is that possible?
- Results are mixed
- Only one architecture used

---

### Official Review · Reviewer_3LH5 · 2024-07-15
**Classification of pediatric Brain Tumors using the fusion of H&E, Ki-67 and GFAP WSI-level features**

**Custom Rating:** 4
**Confidence:** 4

**Review:**

The authors used 3 different WSI stainings (H&E, Ki-67 and GFAP). WSIs were cut into tiles from which features were extracted using UNI-ViT. The features were then fed to a multiple instance learning CLAM model to perform classification. When using multiple image modalities to predict tumor type, the UNI-ViT features were aggregated before applying CLAM model. Using the multi-stain approach improved the differentiation between astrocytoma low grade and high grade. However, the single stain H&E performed better that stain fusion when classifying all 5 tumor types, namely ependymoma, medulloblastoma, ganglioglioma, astrocytoma low grade and astrocytoma high grade.

The idea proposed by the authors of fusing multiple staining types is interesting, the manuscript well written and the methodology clear. However few points are unclear to the reviewer and should be addressed to improve the quality of the manuscript.

2. Data section:
- It is mentioned that some slides/patients were removed from the dataset because of outdated tumor classifications. Does it mean that the remaining patients have tumor classification according to the latest WHO guidelines ? Could it be possible to update the outdated classifications to use the entire dataset ?
- How different are the morphologies of the 5 tumor types on H&E slides ? Is there anything special expected regarding model ability to distinguish certain tumor types ? It is uncleat to the reviewer how the addition of Ki-67 and GFAP should help in the differentiation of the 5 tumor types studied in the paper, is there any rule like Ki-67 pos and GFAP pos = tumor type 1, Ki-67 neg and GFAP pos = tumor type 2,.... ?
- The authors used 256x256 px tiles to extract tiles feature, however it was never mentioned what it the resolution of the dataset (micrometer / pixel)

3. Methodology section:
-The authors mention that they used attention maps to visualize important regions for the classification. These maps were compared to QuPath generated density maps of Ki-67. How was this performed ? What method did the authors used to generate these density maps ? Were these maps then used as ground truth for Ki-67 level ?

4. Results section:
- Last part of the second paragraph the authors write: "Specifically, EP and MED are almost perfectly classified, and ASTR-LGG is correctly classified in most cases. However, ASTR-HGG is incorrectly classified in less than half of the cases, often classified as ASTR-HGG, and GANGvis frequently misclassified as ASTR-LGG.". From the table provided in the manuscript, these results are not visible. For the understanding of the reader, a table giving the results per class or a confusion matrix would be great. Also it would be nice to comment if these results would be expected or not based on each tumor type specificities.

Minor comments:
- p2, last paragraoh from the introduction " enhance predictions and achieve more accurate diagnoses" is written twice.
- "However, ASTR-HGG is incorrectly classified in less than half of the cases, often classified as ASTR-HGG" the reviwer believes that one of the ASTR-HGG should be ASTR-LGG
- Figures 4 & 5: the inclusion of the Ki-67 stained WSI is confusing since no brown regions can be observed at this magnification level. Maybe include a higher magnification crop to show the difference between low/high Ki-67 regions  ?

---

### Decision · Program_Chairs · 2024-07-16

Accept